# LEARNABILITY OF LEARNED NEURAL NETWORKS

## ABSTRACT

This paper explores the simplicity of learned neural networks under various settings: learned on real vs random data, varying size/architecture and using large minibatch size vs small minibatch size. The notion of simplicity used here is that of learnability i.e., how accurately can the prediction function of a neural network be learned from labeled samples from it. While learnability is different from (in fact often higher than) test accuracy, the results herein suggest that there is a strong correlation between small generalization errors and high learnability. This work also shows that there exist significant qualitative differences in shallow networks as compared to popular deep networks. More broadly, this paper extends in a new direction, previous work on understanding the properties of learned neural networks. Our hope is that such an empirical study of understanding learned neural networks might shed light on the right assumptions that can be made for a theoretical study of deep learning.

## 1 INTRODUCTION

Over the last few years neural networks have significantly advanced state of the art on several tasks such as image classification (Russakovsky et al. (2015)), machine translation (Xiong et al. (2016)), structured prediction (Belanger & McCallum (2016)) and so on, and have transformed the areas of computer vision and natural language processing. Despite the success of neural networks in making these advances, the reasons for their success are not well understood. Understanding the performance of neural networks and reasons for their success are major open problems at the moment. Questions about the performance of neural networks can be broadly classified into two groups: i) *optimization* i.e., how are we able to train large neural networks well even though it is NP-hard to do so in the worst case, and ii) *generalization* i.e., how is it that the training error and test error are close to each other for large neural networks where the number of parameters in the network is much larger than the number of training examples (highly overparametrized). This paper explores three aspects of generalization in neural networks.

The first aspect is the performance of neural networks on random training labels. While neural networks generalize well (i.e., training and test error are close to each other) on real datasets even in highly overparametrized settings, Zhang et al. (2017) shows that neural networks are nevertheless capable of achieving zero training error on random training labels. Since any given network will have large error on random test labels, Zhang et al. (2017) concludes that neural networks are indeed capable of poor generalization. However since the labels of the test set are random and completely independent of the training data, this leaves open the question of whether neural networks learn *simple* patterns even on random training data. Indeed the results of Rolnick et al. (2017) establish that even in the presence of massive label noise in the training data, neural networks obtain good test accuracy on real data. This suggests that neural networks might learn some simple patterns even with random training labels. The first question this paper asks is

**(Q1)**: Do neural networks learn simple patterns on random training data?

A second, very curious, aspect about the generalization of neural networks is the observation that increasing the size of a neural network helps in achieving better test error even if a training error of zero has already been achieved (see, e.g., Neyshabur et al. (2014)) i.e., larger neural networks have better generalization error. This is contrary to traditional wisdom in statistical learning theory which holds that larger models give better training error but at the cost of higher generalization error. A recent line of work proposes that the reason for better generalization of larger neural networks is

*implicit regularization*, or in other words larger learned models are *simpler* than smaller learned models. See Neyshabur (2017) for references. The second question this paper asks is

**(Q2)**: Do larger neural networks learn simpler patterns compared to smaller neural networks when trained on real data?

The third aspect about generalization that this paper considers is the widely observed phenomenon that using large minibatches for stochastic gradient descent (SGD) leads to poor generalization LeCun et al..

**(Q3)**: Are neural networks learned with small minibatch sizes simpler compared to those learned with large minibatch sizes?

All the above questions have been looked at from the point of view of flat/sharp minimizers Hochreiter & Schmidhuber (1997). Here flat/sharp corresponds to the curvature of the loss function around the learned neural network. Krueger et al. (2017) for true vs random data, Sagun et al. (2017) for large vs small neural networks and Keskar et al. (2016) for small vs large minibatch training, all look at the sharpness of minimizers in various settings and connect it to the generalization performance of neural networks. While there certainly seems to be a connection between the sharpness of the learned neural network, there is as yet no unambiguous notion of this sharpness to quantify it. See Dinh et al. (2017) for more details.

This paper takes a complementary approach: it looks at the above questions through the lens of *learnability*. Let us say we are considering a multi-class classification problem with $c$ classes and let $\mathcal{D}$ denote a distribution over the inputs $x \sim \mathbb{R}^d$. Given a neural network $\mathcal{N}$, draw $n$ independent samples $x_1^{\mathrm{tr}}, \cdots, x_n^{\mathrm{tr}}$ from $\mathcal{D}$ and train a neural network $\widehat{\mathcal{N}}$ on training data $(x_1^{\mathrm{tr}}, \mathcal{N}(x_1^{\mathrm{tr}})), \cdots, (x_n^{\mathrm{tr}}, \mathcal{N}(x_n^{\mathrm{tr}}))$, where $N(x) \in [c]$ denotes the prediction of $\mathcal{N}$ on $x$. The learnability of a neural network $\mathcal{N}$ is defined to be

$$L(\mathcal{N}) \stackrel{\text{def}}{=} \mathop{\mathbb{E}}_{x^{\mathrm{tr}}, x \sim \mathcal{D}} \left[ \mathbb{1}_{\left\{ \mathcal{N}(x) = \widehat{\mathcal{N}}(x) \right\}} \right] \times 100\%. \tag{1}$$

Note that $L(\mathcal{N})$ implicitly depends on $\mathcal{D}$, the architecture and learning algorithm used to learn $\widehat{\mathcal{N}}$ as well as $n$. This dependence is suppressed in the notation but will be clear from context. Intuitively, larger the $L(\mathcal{N})$, easier it is to learn $\mathcal{N}$ from data. This notion of learnability is not new and is very closely related to probably approximately correct (PAC) learnability Valiant (1984); Kearns & Vazirani (1994). In the context of neural networks, learnability has been well studied from a theoretical point as we discuss briefly in Sec.2. There we also discuss some related empirical results but to the best of our knowledge there has been no work investigating the learnability of neural networks that are encountered in practice.

This paper empirically investigates the learnability of neural networks of varying sizes/architectures and minibatch sizes, learned on real/random data in order to answer (Q1) and (Q2) and (Q3). The main contributions of this paper are as follows:

**Contributions**: The results in this paper suggest that there is a strong correlation between generalizability and learnability of neural networks i.e., neural networks that generalize well are more learnable compared to those that do not generalize well. Our experiments suggest that

- Neural networks do not learn simple patterns on random data.

- Learned neural networks of large size/architectures that achieve higher accuracies are more learnable.

- Neural networks learned using small minibatch sizes are more learnable compared to those learned using large minibatch sizes.

Experiments also suggest that there are qualitative differences between learned shallow networks and deep networks and further investigation is warranted.

**Paper organization**: The paper is organized as follows. Section 2 gives an overview of related work. Section 3 presents the experimental setup and results. Section 5 concludes the paper with some discussion of results and future directions.

## 2 RELATED WORK

Learnability of the concept class of neural networks has been addressed from a theoretical point of view in two recent lines of work. The first line of work shows hardness of learning by exhibiting a distribution and neural net that is hard to learn by certain type of algorithms. We will mention one of the recent results, further information can be obtained from references therein. Song et al. (2017) (see also Shamir (2016); Shalev-Shwartz et al. (2017)) show that there exist families of single hidden layer neural networks of small size that is hard to learn for *statistical query algorithms* (statistical query algorithms Kearns (1998) capture a large class of learning algorithms, in particular, many deep learning algorithms such as SGD). The result holds for log-concave distributions on the input and for a wide class of activation functions. If each sample is used only ones, then the hardness in their result means that the number of samples required is exponentially large. These results do not seem directly applicable to input distributions and networks encountered in practice.

The second line of work shows, under various assumptions on $\mathcal{D}$ and/or $\mathcal{N}$, that the learnability of neural networks is close to 1 Arora et al. (2014); Janzamin et al. (2015); Giryes et al. (2016); Zhong et al. (2017). Recently, Goel & Klivans (2017) give a provably efficient algorithm for learning one hidden layer neural networks consisting of sigmoids. However, their algorithm, which uses the kernel method, is different from the ones used in practice and the output hypothesis is not in the form of a neural network.

Using one neural net to train another has also been used in practice, e.g. Ba & Caurana (2013); Hinton et al. (2015); Urban et al. (2016). The goal in these works is to train a *small* neural net to the data with high accuracy by a process often called *distillation*. To this end, first a large network is trained to high accuracy. Then a smaller network is trained on the original data, but instead of class labels, the training now uses the classification probabilities or related quantities of the large network. Thus the goal of this line of research, while related, is different from our goal.

## 3 EXPERIMENTS

In this section, we will describe our experiments and present results.

### 3.1 EXPERIMENT SETUP

All our experiments were performed on CIFAR-10 Krizhevsky et al. (2009). The $60,000$ training examples were divided into three subsets $D_1$, $D_2$ and $D_3$ with $D_1$ and $D_2$ having 25000 samples each and $D_3$ having 10000 samples. We overload the term $D_i$ to denote both the unlabeled as well as labeled data points in the $i^{\text{th}}$ split; usage will be clear from context. For all the experiments, we use vanilla stochastic gradient descent (SGD) i.e., no momentum parameter, with an initial learning rate of 0.01. We decrease the learning rate by a factor of $\frac{3}{4}$ if there is no decrease in train error for the last 10 epochs. Learning proceeds for $500$ epochs or when the training zero-one error becomes smaller than $1\%$, whichever is earlier. Unless mentioned otherwise, minibatch size of $64$ is used and the final training zero-one error is smaller than $1\%$. For training, we minimize logloss and do not use weight decay. The experimental setup is as follows.

**Step 1** Train a network $N_1$ on (labeled) $D_1$.

**Step 2** Use $N_1$ to predict labels for (unlabeled) $D_2$, denoted by $N_1(D_2)$.

**Step 3** Train another network $N_2$ on the data $(D_2, N_1(D_2))$.

Learnability of a network is computed as $\frac{1}{|D_3|} \sum_{i=1}^{|D_3|} \mathbb{1}_{\{N_1(D_3)=N_2(D_3)\}} \times 100\%$. All the numbers reported here were averaged over 5 independent runs. We now present experimental results aimed at answering **(Q1)**, **(Q2)** and **(Q3)** we raised in Section 1.

### 3.2 EFFECT OF DATA

The first set of experiments are aimed at understanding the effect of data on the simplicity of learned neural networks. We work with three different kinds of data. In this section we vary the data in three ways

- **True data**: Use *labeled* images from CIFAR-10 for $D_1$ in **Step 1**.
- **Random labels**: Use *unlabeled* images from CIFAR-10 for $D_1$ in **Step 1** and assign them random labels uniformly between 1 and 10.
- **Random images**: Use random images and labels in **Step 1**, where each pixel in the image is drawn uniformly from $[-1, 1]$.

For this set of experiments architecture of $N_1$ was the same as that of $N_2$. The networks $N_1$ and $N_2$ were varied over different architectures: VGG Simonyan & Zisserman (2014), GoogleNet Szegedy et al. (2015), ResNet He et al. (2016a), PreActResnet He et al. (2016b) , DPN Chen et al. (2017) and DenseNet Huang et al. (2016). We also do the same experiment on shallow convolutional neural networks with one convolutional layer and one fully connected layer. For the shallow networks, we vary the number of filters in $N_1$ and $N_2$ from $\{16, 32, 64, 128, 256, 512, 1024\}$. We start with 16 filters since that is the minimum number of filters where the training zero one error goes below $1\%$. The learnability values for various networks for true data, random labels and random images are presented in Table 1 for shallow convolutional networks, Table 2 for popular deep convolutional networks and Table 3 for shallow multilayer perceptrons (MLPs).

| Network | Random Labels | Random Images | True Data | True Data Acc. |
|---------|---------------|---------------|-----------|----------------|
| 16 | 21.57±0.66 | 14.19±1.01 | 58.81±0.4 | 53.91±0.56 |
| 32 | 22.07±0.34 | 16.13±0.44 | 68.32±0.3 | 58.67±0.41 |
| 64 | 28.98±0.38 | 22.91±0.65 | 73.04±.25 | 61.37±0.36 |
| 128 | 35.34±0.55 | 31.05±1.75 | 76.12±0.34 | 62.90±0.35 |
| 256 | 40.93±0.62 | 40.5±0.84 | 78.05±0.29 | 63.80±0.32 |
| 512 | 43.51±0.91 | 49.43±2.78 | 79.56±0.5 | 64.43±0.11 |
| 1024 | 46.49±1.15 | 52.06±0.29 | 80.41±.09 | 64.88±0.33 |

Table 1: Learnability comparison of shallow networks on CIFAR-10 dataset with a batchsize of 64 averaged across five independent runs.

| Network | Random Labels | Random Images | True Data | True Data Acc. |
|---------|---------------|---------------|-----------|----------------|
| VGG11 | 17.99±0.34 | 11.30±0.15 | 73.47±0.63 | 72.93±0.36 |
| VGG13 | 16.82±0.54 | 12.51±0.15 | 75.21±1.25 | 75.01±0.56 |
| VGG16 | 17.88±0.49 | 11.97±0.34 | 75.41±0.77 | 75.78±0.60 |
| VGG19 | 17.95±0.62 | 11.78±0.46 | 75.84±0.49 | 76.10±0.28 |
| ResNet18 | 14.96±0.20 | 13.56±0.58 | 69.93±0.74 | 69.98±0.55 |
| Resnet34 | 16.14±0.54 | 13.96±0.07 | 72.22±0.85 | 71.88±0.41 |
| PreActResnet18 | 16.33 ±0.35 | 14.61±0.80 | 70.83±0.75 | 68.35±1.97 |
| DPN26 | 17.03±0.30 | 12.61±0.05 | 70.36±0.70 | 69.84±0.27 |
| DenseNet121 | 19.32±0.26 | 13.35±0.35 | 79.46±0.08 | 79.47±0.67 |
| GoogleNet | 16.10±0.55 | 13.77±0.02 | 78.55±1.26 | 78.58±0.31 |

Table 2: Learnability comparison of popular neural network architectures on CIFAR 10 dataset with a batchsize of 64 averaged across five independent runs.

| Network Depth | Random Labels | Random Images | True Data | True Data Acc. |
|---------------|---------------|---------------|-----------|----------------|
| 1 | 21.39±0.35 | 45.13±0.83 | 50.18±0.72 | 40.88±0.65 |
| 2 | 17.65±0.43 | 34.89±1.29 | 48.58±0.57 | 42.62±0.53 |
| 3 | 15.99±0.21 | 24.78±0.91 | 48.62±1.01 | 42.83±0.63 |
| 4 | 13.74±0.01 | 20.33±0.99 | 48.12±0.83 | 42.74±0.83 |
| 5 | 12.60±0.60 | 16.55±0.12 | 46.12±0.81 | 42.76±0.52 |

Table 3: Learnability comparison of MLPs (Multi Layer Perceptrons) of fixed hidden unit size 64 and varying depth on CIFAR-10 dataset with a batchsize of 64 averaged across five independent runs.

We see from the results that the learnability values of neural networks learned using true data are much larger compared to the values for those learned using random labels or random images. This

| Network | Random Labels | Random Images | True Data | True Data Acc. |
|---------|---------------|---------------|-----------|----------------|
| 16 | 13.52±0.54 | 8.98±0.59 | 38.56±1.10 | 26.13±0.36 |
| 32 | 18.92±0.43 | 16.93±0.90 | 44.05±0.67 | 28.89±0.43 |
| 64 | 22.61±1.13 | 25.74±0.72 | 47.18±0.73 | 30.32±0.18 |
| 128 | 25.88±0.30 | 34.34±2.27 | 48.87±0.47 | 30.71±0.18 |
| 256 | 28.52±0.88 | 43.27±2.15 | 49.75±0.32 | 31.39±0.13 |
| 512 | 29.62±1.07 | 47.45±2.04 | 50.71±0.29 | 31.68±0.19 |
| 1024 | 30.57±1.05 | 48.08±2.41 | 51.06±0.26 | 32.10±0.19 |

Table 4: Learnability comparison of shallow CNNs on CIFAR-100 dataset with a batchsize of 64 averaged across five independent runs.

| Network Arch. \ Class | 0 | 1 | 2 | 3 | 4 | 5 | 6 | 7 | 8 | 9 |
|-----------------------|-----|-----|-----|-----|-----|-----|-----|-----|-----|-----|
| **True Data** | | | | | | | | | | |
| GoogleNet | 10.89 | 8.95 | 9.73 | 8.88 | 12.29 | 9.04 | 9.13 | 11.48 | 8.85 | 10.75 |
| DenseNet | 9.19 | 9.27 | 8.51 | 11.18 | 12.14 | 8.90 | 6.35 | 16.54 | 9.28 | 8.65 |
| Shallow conv 1024 | 10.45 | 9.45 | 9.14 | 9.08 | 9.72 | 12.44 | 10.15 | 10.41 | 9.50 | 9.65 |
| Shallow conv 16 | 11.78 | 9.67 | 9.44 | 9.48 | 9.83 | 10.35 | 9.90 | 10.37 | 9.16 | 10.03 |
| | | | | | | | | | | |
| **Random Labels** | | | | | | | | | | |
| DenseNet | 10.17 | 8.24 | 11.54 | 6.84 | 11.11 | 10.81 | 10.70 | 9.17 | 11.01 | 10.42 |
| GoogleNet | 9.33 | 11.51 | 6.76 | 9.71 | 12.82 | 9.20 | 10.32 | 7.08 | 10.70 | 12.57 |
| Shallow conv 1024 | 11.86 | 10.05 | 13.93 | 10.07 | 9.79 | 12.69 | 14.60 | 4.88 | 4.94 | 7.19 |
| Shallow conv 16 | 8.86 | 9.78 | 11.89 | 10.40 | 10.14 | 7.39 | 10.69 | 10.99 | 12.37 | 7.50 |
| | | | | | | | | | | |
| **Random Images** | | | | | | | | | | |
| DenseNet | 4.05 | 8.07 | 2.02 | 9.02 | 0.49 | 17.59 | 0.75 | 17.85 | 37.46 | 2.70 |
| GoogleNet | 28.78 | 7.29 | 4.56 | 4.50 | 1.13 | 12.12 | 21.79 | 8.45 | 0.93 | 10.46 |
| Shallow conv 1024 | 3.42 | 22.07 | 19.03 | 6.79 | 5.00 | 3.46 | 8.99 | 7.88 | 4.19 | 19.17 |
| Shallow conv 16 | 9.68 | 10.13 | 9.33 | 8.70 | 8.52 | 11.76 | 12.01 | 11.91 | 8.69 | 9.28 |

Table 5: Class wise Percentage distribution for $N_1$ predictions on $D_2$ for CIFAR-10 Dataset. Shallow 16 refers to a single layer ConvNet with 16 number of filters.

clearly demonstrates that the complexity of a learned neural network heavily depends on the training data. Given that complexity of the learned model is closely related to its generalizability, this further supports the view that generalization in neural networks heavily depends on training data. Similar results can be observed for shallow convolutional networks on CIFAR-100 in Table 4.

It is perhaps surprising that the learnability of networks trained on random data is substantially higher than 10% for shallow networks, on the other hand it's close to 10% for deeper networks. Some of this is due to class imbalance: in the case of true data, class imbalance is minimal for all architectures. While, when trained on random labels or random images output of $N_1$ on $D_2$ was skewed. This happened both for shallow networks and deeper networks but was slightly higher for shallow networks. Table 5 presents the percentage of each class in the labels of $N1$ on $D2$. However, we do not have a quantification of how much of learnability in the case of shallow networks arises due to class imbalance and a compelling reason for high learnability of shallow networks.

For any given example, let us denote $\text{TLP}(x) \stackrel{\text{def}}{=} \mathbb{1}_{\{N_1(x)=y(x)\}}$, where $y(x)$ denotes the true label of $x$ and $\text{PLP}(x) \stackrel{\text{def}}{=} \mathbb{1}_{\{N_1(x)=N_2(x)\}}$. Tables 6 and 7 present the percentage of examples for the

| TLP \ PLP | 0 | 1 |
|-----------|-------|-------|
| 0 | 11.68. | 23.38 |
| 1 | 7.93 | 57.01 |

Table 6: $N_1$: Shallow net with 1024 filters, $N_2$: Shallow net with 1024 filters; in percentage

| TLP \ PLP | 0 | 1 |
|-----------|-------|-------|
| 0 | 25.84 | 19.83 |
| 1 | 15.29 | 39.04 |

Table 7: $N_1$: Shallow net with 16 filters, $N_2$: Shallow net with 16 filters; in percentage

| PLP / TLP | 0 | 1 |
|---|---|---|
| 0 | 14.58. | 12.93 |
| 1 | 11.94 | 60.55 |

Table 8: $N_1$: VGG11 and $N_2$: VGG11; in percentage

| PLP / TLP | 0 | 1 |
|---|---|---|
| 0 | 11.21 | 9.97 |
| 1 | 10.23 | 68.59 |

Table 9: $N_1$: GoogleNet, $N_2$: GoogleNet; in percentage

four different possibilities of TLP and PLP for shallow networks while Tables 8 and 9 present these results for VGG-11 and GoogleNet. The key point we would like to point out from these tables is that if we focus on those examples where $N_1$ does not predict the true label correctly i.e., TLP = 0 or the first row in the tables, we see that approximately half of these examples are still learned correctly by $N_2$. Contrast this with the learnability values of $N_1$ learned with random data which are all less than 20%. This suggests that networks learned on true data make simpler predictions even on examples which they misclassify.

### 3.3 Effect of Network Size/Architecture

The second set of experiments are aimed at understanding the effect of network size and architecture on the learnability of the learned neural network. First, we work with shallow convolutional neural networks (CNN) with 1 convolutional layer and 1 fully connected layer.

The results are presented in Table 10. Even though training accuracy is always greater than 99%, test accuracy increases with increase in the size of $N_1$ – Neyshabur et al. (2014) reports similar results for 2-layer multilevel perceptrons (MLP). It is clear that for any fixed $N_2$, the learnability of the learned network increases as the number of filters in $N_1$ increases. This suggests that the larger learned networks are indeed simpler than the smaller learned networks. Note also that for every $N_1$, its learnability values are always larger than its test accuracy when $N_2$ has 16 or more filters. This suggests that $N_2$ learns information about $N_1$ that is not contained in the data.

We performed the same experiment for some popular architectures as in Section 3.2. The results are presented in Table 12. Note that the accuracies reported here are significantly lower than those reported in published literature for the corresponding models; the reason for this is that our data size is essentially cut by half (see Section 3.1). Except for the case where $N_2$ is ResNet18 and $N_1$ is either a VGG or ResNet, there is a positive correlation between test accuracy and learnability i.e., a network with higher test accuracy is more learnable. We do not know the reason for the exception mentioned above. Furthermore, the pattern observed for shallow networks, that learnability is larger than accuracy, does not seem to always hold for these larger networks.

### 3.4 Effect of Batch Size

The third set of experiments are aimed at understanding the effect of minibatch size on the learned model. For this set of experiments, $N_1$ and $N_2$ are again varied over different architectures while keeping the architectures of $N_1$ and $N_2$ same. The minibatch size for training of $N_2$ (**Step 3**) is fixed to 64 while the minibatch size for training of $N_1$ (**Step 1**) is varied over $\{32, 64, 128, 256\}$. Table 13 presents these results. It is clear from these results that for any architecture, increasing the minibatch size leads to a reduction in learnability. This suggests that using a larger minibatch size in SGD leads to a more complex neural network as compared to using a smaller minibatch size.

### 3.5 Variability of predictions

In this section, we will explore a slightly orthogonal question of whether neural networks learned with different random initializations converge to the same neural network, as functions. While there are some existing works e.g., Goodfellow et al. (2014), which explore linear interpolation between the parameters of two learned neural networks with different initializations, we are interested here in understanding if different SGD solutions still correspond to the same function. In order to do this, we compute the confusion matrix for different SGD solutions. If SGD is run $k$ times ($k = 5$ in this case), recall that the $(i, j)$ entry of the confusion matrix, where $1 \leq i, j \leq k$ gives the fraction of examples on which the $i^{th}$ and $j^{th}$ SGD solutions agree. The following are the confusion matrices

| $N_1$ \ $N_2$ | # Params | Test Acc. | 16 | 32 | 64 | 128 | 256 | 512 | 1024 |
|---|---|---|---|---|---|---|---|---|---|
| 16 | 42186 | 53.91±0.56 | 58.81±0.40 | 61.61±0.49 | 63.78±0.54 | 65.09±0.24 | 65.08±0.21 | 65.12±0.19 | 63.94±0.73 |
| 32 | 84362 | 58.67±0.41 | 63.86±0.06 | 68.32±0.30 | 69.25±0.88 | 70.00±1.23 | 70.03±0.62 | 71.14±0.22 | 70.44±0.66 |
| 64 | 168714 | 61.37±0.36 | 67.60±0.04 | 71.37±0.58 | 73.04±0.25 | 74.48±0.05 | 73.80±0.19 | 74.92±0.36 | 74.04±0.01 |
| 128 | 337418 | 62.90±0.35 | 68.80±0.11 | 73.52±0.38 | 75.04±0.10 | 76.12±0.34 | 76.73±0.27 | 76.60±0.14 | 76.89±0.09 |
| 256 | 674826 | 63.80±0.32 | 70.63±0.08 | 74.53±0.22 | 77.53±0.10 | 77.61±0.04 | 78.05±0.29 | 78.10±0.01 | 77.56±0.68 |
| 512 | 1349642 | 64.43±0.11 | 71.69±0.05 | 76.23±0.28 | 77.15±0.05 | 78.07±0.06 | 79.43±0.06 | 79.56±0.50 | 78.96±0.02 |
| 1024 | 2701910 | 64.88±0.33 | 71.85±0.43 | 76.45±0.37 | 78.01±0.32 | 79.19±0.38 | 79.36±0.31 | 79.90±0.19 | 80.41±0.09 |

Table 10: Learnability values for shallow 2-layer CNNs of various sizes. Values in the first column represent the number of filters in $N_1$ and values in the header row represent the number of filters in $N_2$.

| $N_1$ \ $N_2$ | # Params | Test Accuracy | 1 | 2 | 3 | 4 | 5 |
|---|---|---|---|---|---|---|---|
| 1 | 197322 | 40.88±0.65 | 50.18±0.72 | 48.85±1.02 | 48.38±0.32 | 48.04±0.39 | 48.04±0.04 |
| 2 | 201482 | 42.62±0.53 | 48.48±0.10 | 48.58±0.57 | 50.03±0.43 | 48.96±0.33 | 47.80±1.11 |
| 3 | 205642 | 42.83±0.63 | 48.06±0.30 | 48.09±0.19 | 48.62±1.01 | 48.39±1.14 | 47.45±0.24 |
| 4 | 209802 | 42.74±0.83 | 46.35±0.41 | 46.89±0.76 | 48.32±0.95 | 48.12±0.83 | 46.40±0.12 |
| 5 | 213962 | 42.76±0.52 | 44.96±0.92 | 46.14±0.92 | 46.62±0.57 | 46.25±0.61 | 46.12±0.81 |

Table 11: Learnability values for shallow MLPs of various sizes. Values in the first column represent the depth of $N_1$ and values in the header row represent the depth of $N_2$. Each MLP layer had a hidden unit size of 64 followed by a ReLU

| $N_1$ \ $N_2$ | # of Layers, Params | Test Accuracy | VGG11 | ResNet18 | GoogLeNet | DenseNet121 |
|---|---|---|---|---|---|---|
| DPN26 | 89-11574842 | 69.84±0.27 | 68.54±0.97 | 69.84±0.89 | 72.33±0.52 | 72.47±0.15 |
| ResNet18 | 62-11173962 | 69.98±0.55 | 68.79±0.68 | 69.94±0.55 | 71.76±0.24 | 73.33±0.29 |
| ResNet34 | 110-21282122 | 71.88±0.41 | 71.51±0.41 | 71.14±0.01 | 72.22±0.69 | 73.86±0.09 |
| VGG11 | 34-9231114 | 72.93±0.36 | 73.47±0.39 | 69.46±0.57 | 72.39±0.11 | 73.42±0.03 |
| VGG13 | 42-9416010 | 75.01±0.56 | 74.87±0.67 | 70.83±0.08 | 73.84±0.83 | 74.65±0.01 |
| VGG16 | 54-14728266 | 75.78±0.60 | 74.23±1.04 | 72.28±0.14 | 74.40±0.66 | 74.76±0.39 |
| VGG19 | 66-20040522 | 76.10±0.28 | 74.66±0.10 | 71.84±0.15 | 74.29±0.93 | 76.75±0.12 |
| GoogLeNet | 258-6166250 | 78.58±0.31 | 70.75±0.18 | 71.24±1.55 | 78.55±1.58 | 77.74±0.05 |
| DenseNet121 | 362-6956298 | 79.47±0.67 | 73.41±0.37 | 73.95±0.56 | 78.61±0.01 | 79.46±0.01 |

Table 12: Learnability values for various popular architectures. The first column gives the architecture of $N_1$ and the header row shows the architecture of $N_2$. See text for discussion.

| $N_1/N_2$ \ Batch size | 32 | 64 | 128 | 256 |
|---|---|---|---|---|
| VGG11 | 74.56 ±0.71 | 73.75 ±0.07 | 72.05 ±0.83 | 69.10 ±0.53 |
| VGG13 | 74.85 ±0.43 | 73.95 ±1.13 | 73.32 ±0.23 | 69.84 ±0.29 |
| VGG16 | 74.87 ±0.60 | 74.73 ±0.44 | 73.14 ±0.25 | 69.88 ±0.32 |
| VGG19 | 74.44 ±0.40 | 74.38 ±0.03 | 73.03 ±0.15 | 70.74 ±0.52 |
| ResNet18 | 72.51 ±0.87 | 70.06 ±0.45 | 64.98 ±0.17 | 61.72 ±0.64 |
| DenseNet121 | 73.79 ±0.25 | 73.33 ±0.06 | - | - |
| GoogleNet | 72.01 ±0.25 | 71.55 ±0.54 | - | - |

Table 13: Learnability comparison of network architectures on CIFAR-10 dataset with varying batch sizes. For this experiment we fixed N2 to be VGG11 with batch size of 64. GoogleNet and DenseNet architectures ran out of memory for batch size of 128 and 256.

for different SGD solutions, left for a shallow network with 1024 filters and right for VGG-11.

$$
\begin{bmatrix}
1. & 0.93 & 0.93 & 0.93 & 0.93 \\
0.93 & 1. & 0.93 & 0.94 & 0.94 \\
0.93 & 0.93 & 1. & 0.93 & 0.94 \\
0.93 & 0.94 & 0.93 & 1. & 0.93 \\
0.93 & 0.94 & 0.94 & 0.93 & 1.
\end{bmatrix}
\quad
\begin{bmatrix}
1. & 0.73 & 0.73 & 0.73 & 0.72 \\
0.73 & 1. & 0.74 & 0.73 & 0.74 \\
0.73 & 0.74 & 1. & 0.74 & 0.75 \\
0.73 & 0.73 & 0.74 & 1. & 0.74 \\
0.72 & 0.74 & 0.75 & 0.74 & 1.
\end{bmatrix}
$$

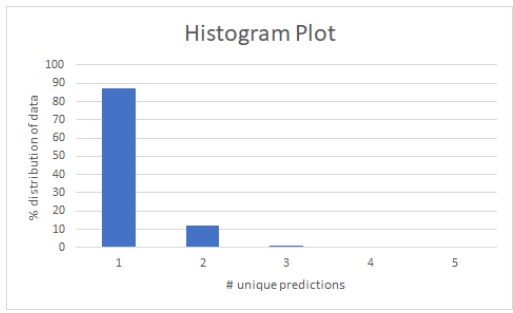 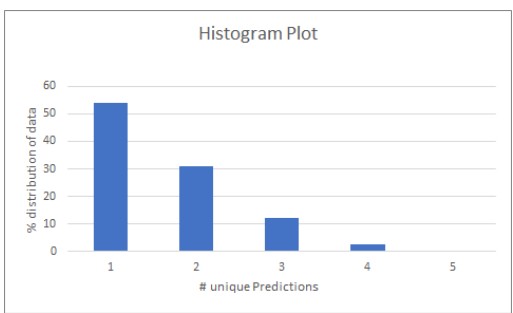

Figure 1: Shallow network with 1024 filters        Figure 2: VGG11

For both the networks, we see that the off-diagonal entries are quite close to each other. This seems to suggest that while the different SGD solutions are not same as functions, they agree on a common subset ($93\%$ for shallow network and $73\%$ for VGG-11) of examples. Furthermore, for VGG-11, the off-diagonal entries are very close to the test accuracy – this behavior of VGG-11 seems common to other popular architectures as well. This seems to suggest that different SGD solutions agree on precisely those examples which they predict correctly, which in turn means that the subset of examples on which different SGD solutions agree with each other are precisely the correctly predicted examples. However this does not seem to be the case. Figures 1 and 2 show the histograms of the number of distinct predictions for shallow network and VGG-11 respectively. For each number $i \in [k]$, it shows the fraction of examples for which the $k$ SGD solutions make exactly $i$ distinct predictions. The number of examples for which there is exactly 1 prediction, or equivalently all the SGD solutions agree is significantly smaller than the test accuracies reported above.

## 4    WHY CORRELATION BETWEEN LEARNABILITY AND GENERALIZATION?

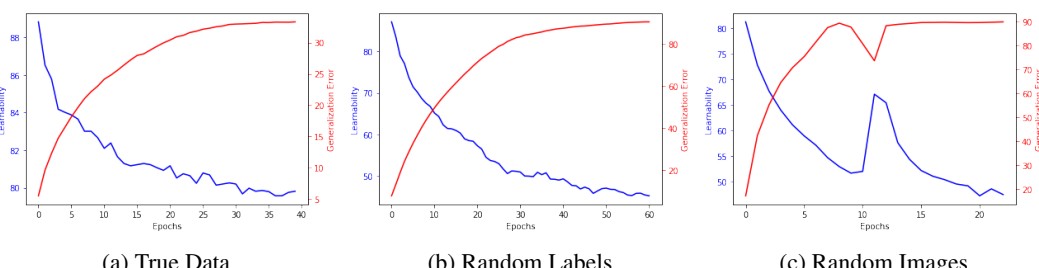

(a) True Data                    (b) Random Labels                    (c) Random Images

Figure 3: Plot of learnability and generalization error vs epochs for shallow 2-layer CNNs

The experimental results so far show a clear correlation between learnability and generalizability of learned neural networks. This naturally leads to the question of why this is the case. We hypothesize that learnability captures the inductive bias of SGD training of neural networks. More precisely, when we start training, intuitively, the initial random network generalizes well (i.e., both train and test errors are high) and is also simple (learnability is high). As SGD changes the network to reduce the training error, it becomes more complex (learnability decreases) and generalization error increases. Figure 3 which shows the plots of learnability and generalizability of shallow 2-layer CNNs supports this hypothesis.

## 5    DISCUSSION AND CONCLUSION

This paper explores the learnability of learned neural networks under various scenarios. The results herein suggest that while learnability is often higher than test accuracy, there is a strong correlation between low generalization error and high learnability of the learned neural networks. This paper also shows that there are some qualitative differences between shallow and popular deep neural

networks. Some questions that this paper raises are the effect of optimization algorithms, hyperparameter selection and initialization schemes on learnability. On the theoretical front, it would be interesting to characterize neural networks that can be learned efficiently via backprop. Given the strong correlation between learnability and generalization, driving the network to converge to learnable networks might help achieve better generalization.

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

# A    LEARNABILITY OF NETWORKS ON MNIST

| No. of Hidden Units | # Params | Learnability | Test Accuracy |
|---|---|---|---|
| 1 | 805 | $97.25 \pm 0.34$ | $37.43 \pm 0.98$ |
| 2 | 1600 | $96.06 \pm 0.51$ | $73.50 \pm 1.49$ |
| 3 | 2395 | $96.65 \pm 0.36$ | $84.58 \pm 0.21$ |
| 4 | 3190 | $95.69 \pm 0.07$ | $89.04 \pm 0.54$ |
| 5 | 3985 | $92.80 \pm 0.74$ | $92.09 \pm 0.26$ |
| 6 | 4780 | $90.27 \pm 2.49$ | $93.16 \pm 0.13$ |

Table 14: Learnability and Accuracy comparison of single layer MLP with varying hidden unit size for $N_1$ on MNIST dataset averaged across five independent runs. For all of the above results we fixed $N_2$ to a single layer MLP with hidden unit size of 4.

In this section, we present in Table 14, the learnability and test accuracy values of single layer MLPs with different number of hidden units trained on MNIST. While we still observe correlation between learnability and test accuracy, the learnability values are much higher than the test accuracy values. This clearly demonstrates that high learnability is does not necessarily require high test accuracy but can occur even when test accuracies are low.

