# OpenReview forum: "Learnability of Learned Neural Networks"
_ICLR.cc/2018/Conference — Reject_

### Official Review · AnonReviewer2 · 2017-11-27
**Very nice paper showing how large networks can actually be "simple", in spite of their large capacity.**

**Rating:** 7
**Confidence:** 4

**Review:**

Summary:
This paper presents very nice experiments comparing the complexity of various different neural networks using the notion of "learnability" --- the learnability of a model (N1) is defined as the "expected agreement" between the output of N1, and the output of another model N2 which has been trained to match N1 (on a dataset of size n).  The paper suggests that the learnability of a model is a good measure of how simple the function learned by that model is --- furthermore, it shows that this notion of learnability correlates well (across extensive experiments) with the test accuracy of the model.

The paper presents a number of interesting results:
1) Larger networks are typically more learnable than smaller ones (typically we think of larger networks as being MORE complicated than smaller networks -- this result suggests that in an important sense, large networks are simpler).
2) Networks trained with random data are significantly less learnable than networks trained on real data.
3) Networks trained on small mini-batches (larger variance SGD updates) are more learnable than those trained on large minibatches.

These results are in line with several of the observations made by Zhang et al (2017), which showed that neural networks are able to both (a) fit random data, and (b) generalize well; these results at first seem to run counter to the ideas from statistical learning theory that models with high capacity (VC dimension, radamacher complexity, etc.) have much weaker generalization guarantees than lower capacity models.  These results suggest that models that have high capacity (by one definition) are also capable of being simple (by another definition).  These results nicely complement the work which studies the "sharpness/curvature" of the local minima found by neural networks, which argue that the minima which generalize better are those with lower curvature.

Review:
Quality:  I found this to be high quality work. The paper presents many results across a variety of network architectures.  One area for improvement is presenting results on larger datasets (currently all experiments are on CIFAR-10), and/or on non-convolutional architectures.  Additionally, a discussion of why learnabiblity might imply low generalization error would have been interesting (the more formal, the better), though it is unclear how difficult this would be.

Clarity:  The paper is written clearly.  A small point: Step 2 in section 3.1 should specify that argmax of N1(D2) is used to generate labels for the training of the second network.  Also, what dataset D_i is used for tables 3-6? Please specify.

Originality: The specific questions tackled in this paper are original (learnability on random vs. real data, large vs. small networks, and large vs. small mini-batch training).  But it is unclear to me exactly how original this use of "learnability" is in evaluating how simple a model is.  It seems to me that this particular use of "learnability" is original, even though PAC learnability was defined a while ago.

Significance:  I find the results in this paper to be quite significant, and to provide a new way of understanding why deep neural networks generalize.  I believe it is important to find new ways of formally defining the "simplicity/capacity" of a model, such that "simpler" models can be proven to have smaller generalization gap (between train and test error) relative to more "complicated" models. It is clear that VC dimension and radamacher complexity alone are not enough to explain the generalization performance of neural networks, and that neural networks with high capacity by these definitions are likely "simple" by other definitions (as we have seen in this paper).  This paper makes an important contribution to this conversation, and could perhaps provide a starting point for theoreticians to better explain why deep networks generalize well.

Pros
- nice experiments, with very interesting results.
- Helps explain one way in which large networks are in fact "simple"

Cons
- The paper does not attempt to relate the notion of learnability to that of generalization performance.  All it says is that these two metrics appear to be well correlated.

---

> ### Author Response · Authors · 2017-12-21
> **Relating learnability and generalization**
>
> Thank you for the review and kind words. The major comment/suggestion is to relate the notion of learnability to that of generalization. Indeed, we have a partial intuitive connection between these two notions based on Figure 3 in the updated draft. We also added Section 4 to discuss this aspect. We hypothesize that learnability captures the inductive bias of SGD training of neural networks. More precisely, when we start training, intuitively, the network is simpler (learnability is high) and generalization error is low (both train and test errors are high). As SGD changes the network to reduce the training error, it becomes more complex (learnability decreases) and the generalization error decreases (train error decreases rapidly while test error does not decrease as rapidly). Training is stopped when the training error becomes less than 1%. At this point, learnability has decreased from its initial high value, and generalization error has increased from its initial low value. Their precise values might be close (as happens in the case of, e.g., N1=N2=VGG11), or not so close (as happens in the case of N1 and N2 being shallow 2-layer CNNs with layer size 1024). Making this connection more formal would be an interesting direction of future work.

---

### Official Review · AnonReviewer3 · 2017-11-27
**This is an interesting empirical attempt at understanding the reproducibility of learned neural networks. Great ideas but needs more work.**

**Rating:** 6
**Confidence:** 4

**Review:**

The proposed approach to figure out what do deep network learn is interesting -- the approach of learning a learned network. Some aspects needs more work to improve the work. The presentation of the results can be improved further.

Firstly, confidence intervals on many experiments are missing (including Tables 3-9). Also, since we are looking at empirically validating the learnability criterion defined by the authors, all the results (the reported confusion tables) need to be tested statistically (to see whether one dominates the other).

What is random label learning of N1 telling us? How different would that be in terms of simply learning random labels on real data directly. Further, the evaluations in Tables 3-6 need more attention since we are interested in the TLP=1 vs. PLP=0 case, and TLP=0 vs. PLP=1 case.

The influence of depth is not clear -- may be it is because of the way results are reported here. A simple figure with increasing layers vs. learnability values would do a better job at conveying the trends.

The evaluations in Section 3.5 are not conclusive? What is the question being tested for here?

What about the influence of number of classes on learnability trends? Some experiments on large class datasets including cifar100 and/or imagenet need to be reported.

--- Comments after response from authors ---

The authors have clarified and shown results for several of the issues I was concerned about. Although it is still unclear what the learnability model is capturing for deeper model or the trends in Section 3.5 (looks like the trends may relate to stability of SGD as well here) -- the proposed ideas are worth discussing. I have appropriately modified my rating.

---

> ### Author Response · Authors · 2017-12-21
> **Response -- part II**
>
> “The evaluations in Section 3.5 are not conclusive? What is the question being tested for here? “
>
> These experiments are an attempt to better understand the notion of learnability as we now explain in a bit more detail than in the paper: While our experiments in previous sections have the learnability values quite concentrated (confidence intervals are small), they say nothing about how concentrated the function computed by N1 itself is across different runs. More precisely, if we train N1 several times using SGD, we expect that the function computed by N1 approximates the data well. However, this function may differ for different runs of SGD and since we are interested in the learnability of the function computed by N1, we would like to understand if it's the same function we are learning each time. In the experiments of this section we are trying to understand the extent to which this happens.
>
> Here is one concrete conclusion of these experiments (also mentioned in the paper). An immediate conjecture suggested by the confusion matrix of VGG11 is that perhaps all that N2 learns is the original data from N1 as the agreement between the functions computed via different SGD runs is approximately the same as the test accuracy (about 73%). This is refuted by Figure 2 as it shows that only on about 55% of data there is full agreement among the different N1's.
>
> Additionally, we can try to relate these experiments to other experiments in the paper: The confusion matrices clearly show that the (function computed by) N1 is considerably more stable in the case of shallow networks than in the case of VGG-11. A similarly stark difference between the two cases is seen also in Tables 7 and 8. In the former, the learnability can be much higher compared to test accuracy; but in the latter, learnability is about the same as test accuracy. It's conceivable that these two phenomena are related and investigating this potential link could provide further insights into both.
>
> Of course, these conclusions lead us to further questions. It is not our claim that we provide a full understanding of learnability and generalization.
>
>
>  “What about the influence of number of classes on learnability trends? Some experiments on large class datasets including cifar100 and/or imagenet need to be reported. “
>
> We have included results on CIFAR100 in Table 4. The results here confirm the trends observed on CIFAR10.

---

> ### Author Response · Authors · 2017-12-21
> **Response -- part I**
>
> “The proposed approach to figure out what do deep network learn is interesting -- the approach of learning a learned network. Some aspects needs more work to improve the work. The presentation of the results can be improved further.
> Firstly, confidence intervals on many experiments are missing (including Tables 3-9). Also, since we are looking at empirically validating the learnability criterion defined by the authors, all the results (the reported confusion tables) need to be tested statistically (to see whether one dominates the other). “
> These were not included in later tables to reduce clutter. We have now included these in the updated version.
>
>  “What is random label learning of N1 telling us? How different would that be in terms of simply learning random labels on real data directly. Further, the evaluations in Tables 3-6 need more attention since we are interested in the TLP=1 vs. PLP=0 case, and TLP=0 vs. PLP=1 case”
>
> Random label learning of N1 (Section 3.2) is trying to answer Q1 posed in the introduction: do neural networks learn simple patterns on random training data? Or equivalently, we could ask: are neural networks learned on random training data simple? The results of Section 3.2 tell us that this is not the case. There is a subtle but substantial difference between learning N2 using data from N1 (which itself is obtained by random label learning, as done in this paper) and learning N2 simply from random labels on real data directly. In the first scenario, the training and test data of N2 are both generated by N1, so it is indeed possible to get even 100% accuracy for N2. On the other hand, in the second scenario, the training and test data for N2 are random and independent. So the test accuracy of N2 will be close to 10% with high probability.
>
> We are sorry, we did not understand your comment about Tables 3-6. Could you please elaborate?
>
>  “The influence of depth is not clear -- may be it is because of the way results are reported here. A simple figure with increasing layers vs. learnability values would do a better job at conveying the trends. “
>
> For clarity, we have now included results on learnability of MLPs with varying depth and a fixed hidden unit size (Table 3). These results suggest that learnability decreases slightly with increasing depth as the number of parameters increase. Note however, that the test accuracies here stay approximately the same with increasing depth. In this case, increasing depth naively does not seem to help.
>
> For popular networks, we need to be careful about drawing conclusions about depth and learnability since a network with higher depth might still have much fewer parameters and hence have low representational power as well as test accuracy. This is the reason we chose to order the networks in increasing order of their test accuracy, which captures their generalizability since all the networks achieve a training error of zero.

---

### Official Review · AnonReviewer1 · 2017-11-28
**The paper poses interesting questions, but learnability doesn't provide many answers**

**Rating:** 4
**Confidence:** 4

**Review:**

Review Summary:
The primary claim that there is "a strong correlation between small generalization errors and high learnability" is correct and supported by evidence, but it doesn't provide much insight for the questions posed at the beginning of the paper or for a general better understanding of theoretical deep learning. In fact the relationship between test accuracy and learnability seems quite obvious, which unfortunately undermines the usefulness of the learnability metric which is used in many experiments in the paper.

For example, consider the results in Table 7. A small network (N1 = 16 neurons) with low test accuracy results in a low learnability, while a large network (N1 = 1024 neurons) gets a higher test accuracy and higher learnability. In this case, the small network can be thought of as applying higher label noise relative to the larger network. Thus it is expected that agreement between N1 and N2 (learnability) will be higher for the larger network, as the predictions of N1 are less noisy. More importantly, this relationship between test accuracy and learnability doesn't answer the original question Q2 posed: "Do larger neural networks learn simpler patterns compared to neural networks when trained on real data". It instead draws some obvious conclusions about noisy labeling of training data.

Other results presented in the paper are puzzling and require further experimentation and discussion, such as the trend that the learnability of shallow networks on random data is much higher than 10%, as discussed at the bottom of page 4. The authors provide some possible reasoning, stating that this strange effect could be due to class imbalance, but it isn't convincing enough.

Other comments:
-Section 3.4 is unrelated to the primary arguments of the paper and seems like a filler.
-Equations should have equation numbers
-Learnability numbers reported in all tables should be between 0-1 per the definition on page 3
-As suggested in the final sentence of the discussion, it would be nice if conclusions drawn from the learnability experiments done in this paper were applied to the design new networks which better generalize

---

> ### Author Response · Authors · 2017-12-21
> **Correlation between test accuracy and learnability "not obvious"**
>
> “-Review Summary: The primary claim that there is a strong correlation between small generalization errors and high learnability" is correct and supported by evidence, . . .. .Do larger neural networks learn simpler patterns compared to neural networks when trained on real data. It instead draws some obvious conclusions about noisy labeling of training data.”
>
> Firstly, we would like to stress that there is “no obvious connection” between test accuracies and learnability. This is clearly demonstrated by a network N1 which predicts the same class (say class 1) for all examples. While N1 is easily learnable, its test accuracy is 10%. The reason for this apparent conflicting behavior is that even though N1 does noisy labeling of training data, the noise introduced is not random – it is highly structured.
>
> The same argument applies to the cases of learned small (N1_small = 16 neurons) and large (N1_large = 1024 neurons) networks. At an intuitive level, one would expect that the noise added by N1_small is much more structured (simpler, smoother) compared to that added by N1_large, since the noise added by N1_small is generated by a small network. In short, higher test accuracy of N1_large does not obviously explain its superior learnability value compared to N1_small. Note that learnability and test accuracy can be substantially different (for shallow networks, the learnability can be up to 16 percent points higher---see Table 7), which shows that N2 learns the structure of N1 apart from learning about noisy version of the original data.
>
> Another way to look at this experiment is to forget that there ever was true data (and hence also forget test accuracies) – all we have are N1_small and N1_large. Given just N1_small and N1_large, considering their relative sizes, traditional wisdom suggests that N1_small is more learnable than N1_large---we think that this has at least as much intuitive force as the hypothesis you suggest. However, that is simply not the case. There is something about N1_large which, despite its large size, makes it much easier to learn than N1_small. This precisely answers Q2: larger neural networks do learn simpler patterns compared to smaller networks when trained on real data.
>
> If you have a look at the included MNIST results in the appendix, we can clearly see that even a very simpler network very few number of parameters and low-test accuracy is highly learnable because of its simplicity.
>
> To sum up, explanation of the correlation between generalizability and learnability does not seem to be obvious. We do offer one partial explanation below in reply to AnonReviewer2.
>
>
>
> “Other results presented in the paper are puzzling and require further experimentation and discussion, such as the trend that the learnability of shallow networks on random data is much higher than 10%, as discussed at the bottom of page 4. The authors provide some possible reasoning, stating that this strange effect could be due to class imbalance, but it isn't convincing enough.”
>
> Following up on your comment, we present the class imbalance values for two deep networks and two shallow networks on true data, random labels and random images in Table 5 in the updated draft. While class imbalance is slightly higher for shallow networks compared to deeper ones on random data, it is indeed the case that the difference in class imbalance is not high. Answering this question does seem to require further experimentation.

---

> ### Author Response · Authors · 2017-12-21
> **Response to other comments**
>
> Other comments:
> “-Section 3.4 is unrelated to the primary arguments of the paper and seems like a filler”
> We think that Section 3.4 perfectly aligns with the theme of the paper i.e., exploring learnability of learned neural networks and its relation to generalization. This section is aimed at answering Q3 posed at the beginning of the paper. It is well-known that networks obtained with higher batch size have poorer generalization. As our experiments indicate, networks trained with higher batch size also have poorer learnability. A priori, it's not clear what to expect from such an experiment on learnability. Thus, our experiments in this section can be thought of another confirmation of our finding that learnability and generalization tend to be correlated.
>
>
> “-Equations should have equation numbers “
> There's only one equation in the paper and it's numbered (1). Did we understand your comment correctly?
>
> “-Learnability numbers reported in all tables should be between 0-1 per the definition on page 3”
> You are correct. The reported values are percent values obtained by multiplying the value in the definition by 100. We have rectified this in the updated version.
>
> “-As suggested in the final sentence of the discussion, it would be nice if conclusions drawn from the learnability experiments done in this paper were applied to the design new networks which better generalize”
>
> One immediate approach to achieve this would be to regularize training so as to guide it towards more learnable networks. Since learnability of a network can be estimated (but this is not very cheap) this is a reasonably concrete approach, though considerable amount of work seems to be required to make this work.
> The final but one sentence of the discussion points out another way for this goal to be achieved: characterizing neural networks that can be efficiently learned via backprop. If such a characterization is available, either regularization of the loss function or modifying the backprop updates might be able to help us design new networks that generalize better.
>
> While training networks with better generalization is certainly a long-term goal of this study, it is outside the scope of current paper. We note that while the concept of flat/sharp minima and its relation to generalization were proposed back in 1997, it took almost 20 years to design a new algorithm (Entropy SGD) that exploits this principle to find networks that generalize better (and is still an ongoing program of work).

---

### Author Response · Authors · 2018-01-05
**New Revision**

After taking the reviewer’s suggestions we have made the following changes:

-	Sec. 3.1: Modified equation (1) to normalize learnability (previous equation multiplied by 100)
-	Sec. 3: Included new learnability results for MLPs (Multi-Layer Perceptrons) and CIFAR 100 dataset
-	All reported tables now have confidence intervals
-	Introduced a table (Table 5 in Sec. 3.2) for showing class-wise percentage distribution for N1
-	Included a new section (Sec. 4) with plots of learnability and generalization error vs epoch
-	Included an appendix about learnability for MNIST

---

### Decision · Program_Chairs · 2018-01-29
**ICLR 2018 Conference Acceptance Decision**

**Decision:**

Reject

**Comment:**

 + The paper proposes an interesting empirical measure of ""learnability"" of a trained network: how well the predictive function it represents can be learned by another network. And shows it empirically seems to correlate with better generalization.
 - The work is purely empirical: it features no theory relating this learnability to generalization
 - Learnability measure is somewhat ad-hoc with moving parts left to be specified (learning network, data splits, ...)
 - as pointed out by a reviewer, learnability doesn't really provide any answers for now.
 - the work would be much stronger if it went beyond a mere correlation study, and if learnability considerations allowed to derive a new approach/regularization scheme that was convincingly shown to improve generalization.